# Automated Student Classroom Behaviors’ Perception and Identification Using Motion Sensors

**DOI:** 10.3390/bioengineering10020127

**Published:** 2023-01-18

**Authors:** Hongmin Wang, Chi Gao, Hong Fu, Christina Zong-Hao Ma, Quan Wang, Ziyu He, Maojun Li

**Affiliations:** 1Department of Mathematics and Information Technology, The Education University of Hong Kong, Hong Kong 999077, China; 2The Key Laboratory of Spectral Imaging Technology, Xi’an Institute of Optics and Precision Mechanics of the Chinese Academy of Sciences, Xi’an 710119, China; 3The University of Chinese Academy of Sciences, Beijing 100049, China; 4Department of Biomedical Engineering, The Hong Kong Polytechnic University, Hong Kong 999077, China; 5School of Information Science and Technology, Northwest University, Xi’an 710127, China

**Keywords:** intelligent system, deep learning, classroom behavior, motion identification

## Abstract

With the rapid development of artificial intelligence technology, the exploration and application in the field of intelligent education has become a research hotspot of increasing concern. In the actual classroom scenarios, students’ classroom behavior is an important factor that directly affects their learning performance. Specifically, students with poor self-management abilities, particularly specific developmental disorders, may face educational and academic difficulties owing to physical or psychological factors. Therefore, the intelligent perception and identification of school-aged children’s classroom behaviors are extremely valuable and significant. The traditional method for identifying students’ classroom behavior relies on statistical surveys conducted by teachers, which incurs problems such as being time-consuming, labor-intensive, privacy-violating, and an inaccurate manual intervention. To address the above-mentioned issues, we constructed a motion sensor-based intelligent system to realize the perception and identification of classroom behavior in the current study. For the acquired sensor signal, we proposed a Voting-Based Dynamic Time Warping algorithm (VB-DTW) in which a voting mechanism is used to compare the similarities between adjacent clips and extract valid action segments. Subsequent experiments have verified that effective signal segments can help improve the accuracy of behavior identification. Furthermore, upon combining with the classroom motion data acquisition system, through the powerful feature extraction ability of the deep learning algorithms, the effectiveness and feasibility are verified from the perspectives of the dimensional signal characteristics and time series separately so as to realize the accurate, non-invasive and intelligent children’s behavior detection. To verify the feasibility of the proposed method, a self-constructed dataset (SCB-13) was collected. Thirteen participants were invited to perform 14 common class behaviors, wearing motion sensors whose data were recorded by a program. In SCB-13, the proposed method achieved 100% identification accuracy. Based on the proposed algorithms, it is possible to provide immediate feedback on students’ classroom performance and help them improve their learning performance while providing an essential reference basis and data support for constructing an intelligent digital education platform.

## 1. Introduction

### 1.1. Background Information on Students’ Classroom Behavior

With the rapid development, penetration, and integration of artificial intelligence technologies in various areas of society, intelligent digital-based education is progressively becoming a hot issue of substantive research [1,2]. Among the many educational research carriers, the intelligent education classroom scenarios are still the commonly adopted educational method [3], which has the outstanding advantages of direct feedback and extensive interaction between teachers and students [4].

Classroom scenarios present complexity and diversity according to different participants and instructional content. Research has shown that in classroom scenarios, students’ classroom behavior is one of the most important factors influencing their academic performance [5]. Compared with high-achieving students, low-achieving students typically spend a significant amount of class time engaged in non-academic work or other academic work [6]. Therefore, investigating student classroom behavior has essential research implications and applicate values for enhancing student performance and promoting instructional strategies [7].

Specifically, the study of classroom behavior through detecting and identifying students’ classroom behavior patterns can provide timely and stage-specific feedback on students’ classroom performance. Effective statistical analysis of students’ behavior patterns will assist students in effectively understanding their learning habits, timely correcting their poor classroom behavior, improving learning strategies, adjusting learning progress, and deepening their understanding and absorption of knowledge.

Furthermore, the analysis of students’ classroom behavior is especially beneficial for students with special education needs (SEN) and developmental disabilities, such as attention deficit and hyperactivity disorder (ADHD) [8], autism spectrum disorder (ASD) [9], and learning disabilities [9,10]. Conducting classroom behavior analysis is crucial to improving these students’ classroom performance and enhancing their classroom concentration. The percentage of school-aged children diagnosed with developmental disorders is increasing dramatically each year due to various environmental factors such as location, level of education, and medical care. In addition, the percentage of children with developmental disorders increased to 17.8% of all children (3–17 years old, the United States). The proportion is substantial, with approximately one in six children diagnosed with a disease [11]. Specifically, ADHD has the broadest range of effects on all developmental disorders and has the most significant prevalence among children. Characteristics of children with ADHD include inattention, hyperactivity, and impulsivity. Students with developmental disorders generally suffer from academic problems due to physical or psychological issues, and their classroom performance is difficult to self-control.

The study of classroom behaviors of students with developmental disabilities can be used to detect and identify their classroom behaviors automatically to a large extent. It can help them improve their self-awareness, enhance their concentration, and effectively achieve supplementary education without external interventions [12]. Auxiliary education based on non-artificial reminders can greatly relieve their learning pressure, ease learning difficulties and anxiety, increase knowledge and improve environmental adaptability, and promote a virtuous cycle of learning [13].

Finally, due to the need to build intelligent digital education platforms for schools and parents, the study of classroom behavior can further refine students’ learning performance at school [14], optimize school teaching services, improve teaching strategies, and facilitate communication and exchange among multiple parties [15]. The intelligent digital platform is designed with students as the primary body and their classroom behaviors as the principal way of measuring their classroom status in order to enhance students’ learning performance and optimize the teaching services of teachers. The perception and identification of students’ classroom behaviors open the door to the development of an intelligent digital education platform.

### 1.2. Literature Review

In this part, we review the literature from the perspectives of ‘Existing methods and the limitations’ and ‘Advanced methods on human activity recognition’ to demonstrate the previous work on the perception and identification of student classroom behaviors.

#### 1.2.1. Existing Methods and the Limitations

The previous research on students’ classroom behavior in the traditional education field is often based on statistical survey methods, requiring teachers to observe the classroom behavior of the entire class or a smaller number of people as an observer over a period within the classroom and to record their behavior [16]. In these circumstances, the teacher plays the role of the evaluator to assess the student’s behavior patterns. This manual, vision-based approach is usually surpassed in identifying inappropriate classroom activities, but the teacher’s one-to-many nature at the time of the count results in the poor perception of the finer classroom behaviors of most students [17]. Furthermore, this visual-based artificial approach to behavior analysis is undoubtedly time-consuming, labor-intensive, and highly subjective. It is highly likely to violate students’ privacy through external interventions and create learning ancients. It cannot give objective science-based judgments, thus preventing a comprehensive classroom behavior assessment. For school-age children with developmental disabilities, classroom behavior management and interventions are the primary methods for improving their classroom performance. There are three common primary types of classroom behavior interventions: one-on-one peer help or parent coaching [18], instructional task modification [19], and self-monitoring [20]. However, while this traditional intervention method has helped children with developmental disabilities improve their classroom task completion rate and classroom behavioral performance, this behavioral intervention undoubtedly consumes many resources in terms of monitoring children’s classroom behavior. It requires significant human and material resources to assist children’s classroom learning process.

#### 1.2.2. Advanced Methods on Human Activity Recognition

With the development and popularization of artificial intelligence technologies, research on scenario-based understanding and behavioral analysis has shone in practical application scenarios [21]. With the help of data-driven and algorithmic reasoning, machine learning theory provides the feasibility of achieving a one-to-one accurate understanding and assessment of students’ classroom behaviors [22], especially for fine-grained behavioral analysis that is difficult to be taken into account by manual statistics. Some scholars have already implemented AI techniques with classroom scene understanding and achieved better results. For example, intelligent classroom systems that assist teachers in teaching and personalize students’ learning by building front-end interactive learning environments for teachers and students and back-end intelligent learning management systems [23]. Adaptive education platforms that solve students’ specific learning problems provide personalized teaching and improve students’ learning experiences according to their needs and their abilities [24]. However, little research has been conducted on AI-based classroom behavioral understanding due to the immature combination of technologies and the niche nature of the educational scenario, making it almost impossible to find corresponding work for reference. Although classroom behavioral activities are too complex and refined, it still belongs to the domain of behavior recognition, so we can help build an intelligent classroom-acceptable behavior perception system by referring to the relevant theories of human activity recognition. A brief overview of the human activity recognition approach is presented in the following section.

The mainstream approaches for human activity recognition can be roughly classified into vision-based and sensor-based based on different data sources [25]. Vision-based behavioral analysis systems usually use single or multiple RGB or depth cameras to collect images or video data of participants’ behavioral information, environment, and background information in a specific activity space [26]. Moreover, after feature extraction of the collected data through image processing techniques and computer vision methods, they can be used to identify participants’ behavior through algorithm learning and inference. Research conducted by numerous scholars applying vision methods in the field of human activity recognition includes: identifying group behavior and classifying abnormal activities in crowded scenes for surveillance as well as public personal safety purposes [27,28,29], including analysis of fall detection, patient monitoring and other behavioral recognition of individuals to improve the quality of human life through vision [30,31]. However, since the vision-based data acquisition equipment for human activity behavior recognition mainly relies on cameras, it is vulnerable to environmental conditions such as light and weather, the shooting range and angle, and a large amount of acquired data storage. The reference of participants’ activity is easily affected by environmental occlusion and privacy issues. Due to the influence of these factors, vision-based behavior analysis systems have not yet been widely used. In contrast, sensors have the advantages of high sensitivity, small size, accessible storage, and wide applicability to various scenarios, which can avoid various problems in using vision devices, so they are now widely embedded in mobile phones, smartwatches and bracelets, eye-tracking devices, virtual/augmented reality headsets, and various intelligent IoT devices [32]. Meanwhile, along with the widespread popularity of mobile Internet and the increasing demand for daily public use of intelligent devices, the problems of inconvenience in carrying and limited endurance of traditional sensor-based devices have been effectively solved in various application scenarios, and they have now become one of the mainstream methods for human activity recognition [25]. Scholars have applied sensing devices for intelligent activity recognition in several daily domains: Alani et al. achieved 93.67% accuracy in 2020 using a deep learning approach to recognize twenty everyday human activities in intelligent homes [33]; Kavuncuoğlu et al. used only a waist sensor to achieve accurate monitoring of fall and daily motion data, achieving 99.96% accuracy in 2520 data [34].

### 1.3. Contributions and Structure

The contributions of this paper include: 1. Artificial intelligent based behavior recognition is applied to the classroom environment for the first time, and an intelligent system with motion sensors to perceive and identify classroom behavior is built. 2. Based on sensor hardware devices, a classroom behavior database (SCB-13) including 14 common classroom behaviors collected from 13 participants is constructed. 3. A method of extracting valid sensor data segments based on an improved Voting-Based Dynamic Time Warping algorithm (VB-DTW) is proposed. 4. An intelligent identification method is proposed to recognize 14 common classroom behaviors based on valid behavior segments combined with a 1DCNN algorithm, and the proposed method achieved 100% recognition accuracy on a self-constructed dataset (SCB-13).

The second part of this paper describes the data hardware acquisition system and the relevant characteristics of the data; the third part gives a brief overview of the basic principles of the algorithm; the fourth part is the experimental results and comparative analysis; finally, the paper provides a conclusion.

## 2. Materials and Methods

### 2.1. Participants

In this study, we recruited 13 participants to carry out a feasibility study on the possibility of accurately identifying students’ classroom behavior. The participants, aged from 20 to 26 years, were invited to participate in a classroom behavioral simulation experiment. This population consisted of 6 males and 7 females without special educational needs or developmental problems. They were culturally literate and able to comprehend, imitate, and model classroom behaviors accurately. Participants signed consent forms approved by the Ethics Committee of The Education University of Hong Kong (Approval Number: 2021-2022-0417) before data collection.

### 2.2. Experimental Design

For each participant, 5 sets of experimental data were gathered, and a total of 65 sets of data were collected. In each trial, participants were tasked with simulating 14 common classroom behaviors, and Table 1 shows the design of each motion. Each motion lasts for 20 s, which can be divided into the valid time duration doing the motions and the sitting still time. Except for when motion happens, the rest of the period is referred to as sitting still time.

The actual hardware system used in the acquisition system is MPU6050, the main hardware processing chip is ESP-8266, the acquisition data bit rate is 115,200 Hz, the Arduino hardware platform is used for programming control, and the sensor data is stored in a .CSV file format via the computer’s USB port using Python program. Figure 1a illustrates the schematic diagram of the 3D acquisition system, and 3 cameras are respectively installed on the participant’s left side, front side, and diagonal rear to record the participant’s vision motion data. As depicted in Figure 1b, in order to investigate the effect of the sensor in different positions, the sensors are positioned in the middle of the spine and the right shoulder of the participant. In addition to the 14 motions, there is a 20-s system calibration time at the beginning of the experiment to reduce the initial error caused during data acquisition, and the total duration of each experiment is 5 min. The sensors generate 7 channels of data: accelerometer (*x*-axis, *y*-axis, and *z*-axis) data, gyroscope (*x*-axis, *y*-axis, *z*-axis) data, and temperature data. The participants’ motion information can be measured using accelerometer data in various directions. Gyroscope data can monitor angular velocity to determine an object’s position and rotational orientation. Due to their susceptibility to environmental factors, temperature data are insufficient for use in a motion recognition system.

### 2.3. Experiment Data Introduction

SCB-13: The self-built dataset SCB-13 of this paper is made up of the above 13 participants’ classroom behavior sensor data. The dataset will be used for later data analysis and model accuracy testing. Furthermore, we intend to provide a brief explanation of the experiment data of the back sensor from the perspective of an intuitive explanation.

#### 2.3.1. Multiple Channel Data Display

After separating the gathered data by 14 given motion patterns, the 65 sets of data for the same motion are averaged to eliminate individual motion differences. Figure 2 displays the processed 6-channel data when motion 6 (raising hand while standing up) is selected as a sample motion. The data demonstrate that motion occurrence and stable state can be acquired during the valid duration of motion and sitting still time, respectively. Notably, when the students get up and raise their hands, the *Z*-axis data of the accelerometer change the most, which is consistent with the actual situation. It confirms the viability of using sensors for behavior identification.

#### 2.3.2. Display of Different Motions of the Same Participant

A participant was selected randomly, and his/her 4 common classroom behaviors (motion 1 sitting still, motion 5 turning around and looking around, motion 6 raising hand while standing up, and motion 8 standing up and sitting down) were displayed in the accelerometer(acc) channel and the gyroscope(ypr) channel, as shown in Figure 3. There are observable changes in the data between different motions of the same volunteer. Nonetheless, the data patterns of motions 6 and 8 are comparable to some extent, providing the classification a challenging problem.

#### 2.3.3. Display of Different Participants with the Same Motion

Figure 4 shows the accelerometer data and gyroscope data of motion 6, raising hands while standing up, which were collected from 4 randomly picked individuals in order to display the differences in motion between various participants.

The preceding diagram demonstrates that various participants have distinct motion pattern characteristics, even for identical motion. It may be caused by variances in personal posture and habitual behaviors. This necessitates that the established model has robust generalization performance, capable of identifying the distinctions between the characteristics of various motion patterns while allowing for modest variations within the same motion. A comparison of the accelerometer and gyroscope data determined that the gyroscope data has more complex properties and fewer noise points, making it more ideal for the learning and reasoning of the neural network. Before generating the network’s standard input, it is necessary to address the extraction and separation of valid data segments since the same motion of different participants occurs at different times and lasts varying times. Taking into account the temporal features of the data, we attempt to extract valid segments of the entire motion time in this article, which was detailed demonstrated in the identification algorithm.

### 2.4. Identification Algorithm

Overall, the algorithm is divided into 3 stages: the extraction of valid segments based on the Dynamic Time Warping algorithm, data augmentation, and a deep learning-based classification algorithm. The whole process of the algorithm can be shown in Figure 5. Further, about the classification algorithm, we picked the most typical Deep Neural Networks (DNNs) as the classification benchmark and investigated the classification accuracy of the RNN-based method and the CNN-based method to explore the impact of various algorithms on the precise perception and identification of classroom behaviors.

#### 2.4.1. Voting-Based DTW (VB-DTW) Valid Segment Extraction Algorithm

Initially, we normalized the collected data to eliminate large differences in data values, which can hinder the convergence of the model. We scale the features contained in each channel by the maximum value and minimum value to the interval [0, 1] without affecting the numerical distribution:(1)$$X=\frac{X-Xmin}{Xmax-Xmin}$$

Developing a distinctive and suitable method for feature representation is necessary in order to assess if motions can be accurately distinguished from the continuous and substantial stream of sensor data. The classification accuracy is determined by the algorithm’s capacity to accurately extract the features in each motion sequence, particularly for sequences having temporal properties. Even though each motion’s recommended acquisition time is equal, the valid duration of each motion varies due to participant differences during the acquisition process. The ratio of the motion’s valid segment to its total time segment is insufficient for some motions (such as raising a hand on a seat, standing up and raising a hand, standing up and sitting down, and knocking on a table), making it challenging to identify motion patterns and represent motion features. To accurately identify the motion mode of each motion, we must differentiate the sitting still state from the valid duration data. In this context, we proposed an improved algorithm for signal extraction based on the Dynamic Time Warping (DTW) algorithm [35], which names as Voting-Based DTW (VB-DTW) valid segment extraction algorithm.

Since the valid motion segments are surrounded by the “sitting still” data in this work, we must divide the raw motion data into tiny sequences to efficiently locate the valid segment rather than process an entire motion segment directly. To extract the valid segments, we divide the raw motion data with a length of 50, which splits the entire 2000 motion data into 40 smaller slices. Utilizing the VB-DTW algorithm, we figure out the minimum warped path of 2 adjacent slices, a total of 39 warped path values yields for each motion from a total of 40 slices. The average warped path value of the motion is utilized as the threshold, and the combined vote of 4 neighboring warped paths is used to evaluate if the slices correspond to valid motion clips. The effective segment length of the final motion is determined by connecting the extracted valid segments. In addition, to address the issue of varied lengths for each extracted valid motion segment, we uniformly downsample the extracted valid motion segments to 285 in order to make model training easier. We apply the VB-DTW algorithm on the remaining thirteen types of motions except the sitting still since the complete motion sequence of the sitting still is a valid segment of the motion. As a result, we directly downsample the sitting still data to a length of 285. The whole process of the VB-DTW-extracted valid segment algorithm can be shown in Algorithm 1.
**Algorithm 1: Voting-Based DTW (VB-DTW) valid segment extraction algorithm****Input:**$\mathrm{Segments}\;\mathrm{of}\;\mathrm{original}\;\mathrm{data}:\;S_{i},\;i=1,2,3,\ldots ,N,\;where\;N=40$;$\mathrm{For}\;\mathrm{each}\;\mathrm{time}\;\mathrm{sequence}:\;S_{i}=S_{i}^{1},\;S_{i}^{2},\;S_{i}^{3},\;\ldots ,S_{i}^{M},\;where\;M=50$.**Initialization:**Dv={ }, Sv={ }voting set = $\left\{D_{j-2},\;D_{j-1},\;D_{j},D_{j+1},D_{j+2}\right\}$
1: **while** 1≤i≤N−1 **do**2:     $D_{i}\leftarrow DTW\left(S_{i},S_{i+1}\right)$
3: **end while**4:$\;threshold\leftarrow \frac{1}{N-1}\overset{N-1}{\underset{i=1}{\sum }}D_{i}$
5: **while** 3≤j≤N−2 **do**6:       **for** $D_{value}\;in\;\;voting\;set$ **do**7:            count ← 08:            **if** $D_{value}>threshold$ **then**9:                  count←count+1
10:  if count ≥3 **then**11:        Dv←Dv ∪ j12:   j; j←j+1
13: **end while**14: **for** j in  {1,2,N−1,N} **do**15:       **if** $D_{j}>$ **then**16:            Dv←Dv ∪ j
17: $Sv\leftarrow \;Sv\;\cup \;\left\{S_{D_{v}^{k}},S_{D_{v}^{k}+1}\right\}\;\left(k=1,2,3,\ldots length\left(Dv\right)\right)$
**Output:**$\mathrm{DTW}\;\mathrm{value}\;\mathrm{of}\;\mathrm{time}\;\mathrm{series}:\;D_{j},\;j=1,2,3,\ldots ,N-1$;$\mathrm{Valid}\;\mathrm{segment}\;\mathrm{slices}:\;S_{v},\;\;S_{v}\in S_{i}$.

#### 2.4.2. Data Augmentation

Data augmentation assists in resolving the overfitting issue caused by insufficient data sets during model training. Contrary to the data augmentation methods for image data, time series data augmentation confronts several formidable obstacles, including 1. the fundamental features of time series sequences are underutilized, 2. different jobs necessitate the use of distinct data augmentation techniques, and 3. the issue of sample category imbalance.

Traditional time series data augmentation methods can be subdivided into time domain-based data enhancement to convert original data or to inject noise; frequency domain-based data enhancement converts data from the time domain to the frequency domain and then applies enhancement algorithms; and simultaneous time domain and frequency domain analysis. To prevent the issue of model overfitting caused by insufficient data, to strengthen the model’s robustness, and to generate a high number of data samples, we use the window slicing-based method as the data enhancement technique. Window slicing separates the original data of length n into n − s + 1 slices with the same label as the raw segment, using S as the new slice length. During the training process, each slice is sent to the network independently as a training instance for prediction. During testing, the separated slices are also submitted to the network, and the majority vote is utilized to determine the original segment’s label. In this model, we select a slice length of 256, which corresponds to approximately 90% of the original length of 285. Figure 6 depicts the data augmentation method, which divides the down-sampled valid motion sequence into 30 new slices.

#### 2.4.3. Deep Learning-Based Classification Algorithm

We explored 2 categories, Recurrent Neural Networks (RNN) based methods and Convolutional Neural Network (CNN) based methods. The RNN-based method tries to represent data attributes based on temporal properties. Long Short Term Memory network (LSTM) [36] and Bidirectional Long Short Term Memory network (BiLSTM) [37] are the specific algorithms chosen for RNN-based methods. The CNN-based method can extract features by performing convolution on the data and focusing on the data’s spatial characteristics. The chosen method for CNN-based methods is 1DCNN [38]. The basic deep neural network (DNN) is chosen as a simple benchmark model that aims to evaluate the performance of various algorithms from these 2 categories. The reason for comparing these four models is that this paper aimed to explore the more classical, advanced, and effective models of temporal data processing for the performance of perception and identification of students’ classroom behavior tasks. The choice of these four classical models helped us to achieve the goal of presenting the best results of our model compared to the rest of the models.

(1)LSTM and BiLSTM

Recurrent neural networks (RNNs) are uniquely valuable compared to other neural networks for processing interdependent sequential data, such as text analysis, speech recognition, and machine translation. It is also widely used in the field of sensor-based motion recognition due to its property of recursion in the direction of sequence evolution, and all recurrent units are linked in a chain [39].

However, the conventional RNN has a short-term memory problem because the RNN cannot memorize and process more comprehensive sequence information, as the layers in the pre-recursive stage will stop learning due to the vanishing gradient problem or exploding gradient problem caused by backpropagation. For the problem that the later data input has more influence and the earlier data input has less influence on RNN, in 1997, Hochreiter and Schmidhuber proposed the Long Short Term Memory Network (LSTM), which successfully solved the limitation of RNN in processing long sequence data and was able to learn the long-term dependence of sequence data features. LSTM proposed the internal mechanism of ‘gates’ used to regulate the flow of feature information, including input gates that control the reading of data into the unit, output gates that control the output entries of the unit, and forgetting gates that reset the contents of the unit. The specific LSTM structure is shown in Figure 7, and a new vector C representing the cell state is added to the LSTM.

Both traditional RNN and LSTM can only predict the output of the next moment based on the information of the previous moment. While in practical applications, the information of the next moment may also have a significant influence on the output state of this moment. Bi-directional LSTM (Bi-LSTM) combines 2 traditional LSTM models and uses 1 of them for forward input and the other for reverse input to fuse the information of the previous and subsequent moments for inference. Its structure is shown in Figure 8.

(2)1DCNN

One-dimensional convolutional neural networks (1DCNN) have strong advantages for sequence data because of the powerful ability to extract features from fixed-length segments in 1-dimensional signals. Also, the adaptive 1DCNN only performs linear 1D convolutions (scalar multiplication and addition), thus providing the possibility of real-time and low-cost intelligent control over hardware [40]. The basic structure of 1DCNN is shown in Figure 9. The kernel moves on the sequence data along the time axis to complete the feature extraction of the original data.

In conclusion, the algorithm utilized the VB-DTW algorithm to extract valid segments, and then window slicing was used to augment the data and achieve a 30-times dataset increase. For classification, we employ 2 categories of networks. For the RNN-based method, the LSTM network and Bi-LSTM network are chosen, as well as the 1DCNN for the CNN-based method. These 2 different types of networks’ abilities and contributions to percept and identify students’ classroom behavior are assessed.

#### 2.4.4. Evaluation Metrics

(1)Valid Segments Extraction

In order to demonstrate the accuracy of the valid segments obtained by the VB-DTW algorithm, we hand-crafted labeled the indices of all valid motion segments as the benchmark. We measure the similarity between the index of extracted data slices (represented as A) and the benchmark (represented as B) using the Jaccard index. The Jaccard index is used to determine the degree of similarity between limited sample data and is defined as the sample intersection size divided by the sample union size. The equation is:(2)$$J\left(A,\;B\right)=\frac{\left|A\;\cap B\right|}{\left|A\;\cup \;B\right|}=\frac{\left|A\;\cap B\right|}{\left|A\right|+\left|B\right|-\left|A\;\cap B\right|}$$

(2)Motion Identification

In order to verify the classification performance of the model, we usually use the accuracy rate to characterize it, that is, the proportion of the number of samples with accurate classification (represented as a) to the total number of samples (represented as m) of this type. Expressed by the following formula:(3)$$accuracy=\frac{a}{m}$$

## 3. Results

In summary, based on the need to understand the classroom behaviors of school children in educational scenarios, sensor-based devices provide an effective way to identify classroom behaviors intelligently. Therefore, this paper proposes the VB-DTW algorithm based on wearable sensors combined with artificial intelligence technology to achieve intelligent recognition of school children’s classroom behaviors. Based on the recognition results, it is possible to provide immediate feedback on students’ classroom performance and help them improve their learning performance while providing an essential reference basis and data support for constructing an intelligent digital education platform.

### 3.1. Identification Algorithm Valid Segmentation Results

For the 65 groups of motions with the same label, we calculate the Jaccard index of each channel of acc and ypr and then determine the average Jaccard index for each motion by averaging the six-channel values. As shown in Table 2, all the extracted valid segments’ indices except lying on the desktop and writing notes are more than 88% similar to the benchmark. The Jaccard index of lying on the desktop and writing notes is worse than other motions, which may be due to the sensor data not changing significantly during motion times, as well as the warped path between the adjacent paths being near. This is a weakness in our proposed VB-DTW algorithm, which makes the algorithm inefficient for long-term recognition of a substantial portion of near-static data. We will continue to investigate the most effective approach to dealing with precise and effective segment extraction in subsequent tests.

### 3.2. Motion Identification Results

Furthermore, the performance of the aforementioned four models in accurately classifying classroom behavior is evaluated in order to measure the influence of different classification models on the self-constructed dataset (SCB-13). A deep neural network (DNN) is chosen as a simple benchmark model for the purpose of evaluating the efficacy of various algorithms. Separately for the back sensor and shoulder sensor, the research tests the accelerometer data (acc), gyroscope data (ypr), and accelerometer and gyroscope data (acc + ypr). The research confirms the effect of classifying sensor data using LSTM and BiLSTM networks, respectively, taking into account the time-series characteristic of the data. In addition, from the perspective of one-dimensional signal feature extraction, the research uses 1DCNN to extract and classify data features in a more “intelligent” mode. The results of the experiments carried out are listed in Table 3 below.

Based on a comprehensive evaluation of the experiment outcomes, we have determined that both DNN and LSTM networks are generally useful in distinguishing classroom behaviors from the three channels’ data of the accelerometer or gyroscope. However, when accelerometer and gyroscope data are incorporated into the network input, the classification effect of the DNN and LSTM network is significantly enhanced, demonstrating that more data channels are beneficial for the expression and differentiation of features.

The main experiment results show that, compared to DNN and LSTM networks, the BiLSTM network significantly improves the identification accuracy of classroom behavior. In addition, BiLSTM networks are capable of a more robust feature representation, whether for three-channel data (accelerometer, gyroscope) or six-channel data (accelerometer and gyroscope), demonstrating that the combination of forward-backward LSTM neural network for the learning of feature representation has been significantly improved.

Compared to the other three networks, the unique and potent feature extraction capabilities for sequence data demonstrated by the 1DCNN network stands out. Combining accelerometer and gyroscope data, the 1DCNN achieves classification accuracy of 100% and 98.8% for the back and shoulder sensors, respectively. In terms of model complexity and computing speed, 1DCNN is considerably superior to LSTM and BiLSTM.

In general, the data collected by the back sensor is more stable than that collected by the shoulder sensor, allowing for the differentiation of classroom activities on a wider scale. For motion classification, the gyroscope is superior to the accelerometer, despite neither being as accurate as when accelerometer and gyroscope data are used simultaneously in the classification.

## 4. Discussion

### 4.1. Ablation Study

#### 4.1.1. Effect of VB-DTW Valid Segment Extraction

To evaluate the effectiveness of the proposed VB-DTW algorithm for valid segment extraction, we chose the data with the best classification impact (the combination of acc and ypr data) to investigate how valid segment extraction affected the action classification results. Table 4 displays the test results. According to the test results, it can be inferred that the results with VB-DTW valid segment extraction generally have higher accuracy than those without VB-DTW. The 1DCNN model outperforms the other algorithms in terms of classification accuracy for valid segment extraction.

#### 4.1.2. Effect of VB-DTW Augmentation

In order to compare the accuracy of the model with and without data augmentation, we still select the data (the combination of Acc and Ypr data) with the highest level of classification accuracy. Table 5 displays the test results. The test results show that the model’s classification accuracy with and without data augmentation is significantly different, and the special benefits of 1DCNN in the categorization of time series data are not reflected. These results might be brought on by the issue of data overfitting by the insufficient amount of data we gathered. As a result, for datasets with fewer data, the proposed algorithm needs to apply data augmentation on the dataset.

According to the results and discussions, the proposed VB-DTW algorithm, based on wearable sensors and artificial intelligence technology, achieves intelligent perception and identification of school-aged students’ classroom behaviors. Furthermore, effective, valid segment extraction methods, as well as data augmentation in model design, are essential for the network’s superior performance. Intelligent recognition of school-age children’s classroom behavior can provide timely feedback, allowing the children, particularly those with special education needs, to grasp their classroom behavior in real-time and obtain assistance in the classroom without being labor-intensive.

### 4.2. Limitation of the Proposed Method

However, the proposed method has several limitations, particularly when students’ classroom behaviors do not change significantly over time (e.g., writing notes). The proposed method cannot efficiently extract the segments of students’ motions. This issue happened because the segments could not be extracted successfully due to the warped path of the DTW algorithm between adjacent paths being near since the absence of significant changes in the sensor data during motion. As a result, the proposed VB-DTW algorithm is inefficient for the long-term recognition of the majority of near-static data. In future work, we will still explore the most efficient way of dealing with precise and valid segment extraction.

## 5. Conclusions

The purpose of this paper is to provide auxiliary education by intelligently perceiving the behavior of students during classroom scenarios by integrating sensor equipment with AI technology. In this article, an improved algorithm which was named VB-DTW is proposed for separating valid sensor signals based on the DTW algorithm, and the effectiveness is validated using the Jaccard index. It provides the capacity to discern accurately between static and dynamic data. In addition, four classical deep learning network structures are compared for the accuracy of classroom behavior classification. It is discovered that the 1DCNN algorithm has the highest accuracy rate, particularly when accelerometer and gyroscope data are aggregated, where the recognition accuracy rate reaches 100%. We anticipate classifying more classroom activities based on hardware in real time and achieving multi-modal identification by fusing sensor data and visual data in future studies.

## Figures and Tables

**Figure 1 bioengineering-10-00127-f001:**
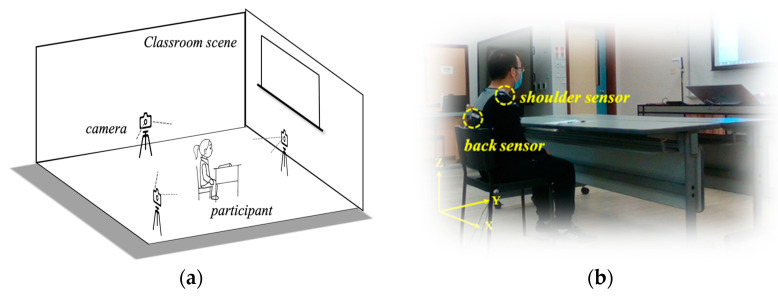
The acquisition system of the experiment. (**a**) The schematic diagram of the motion acquisition system in the classroom scene; (**b**) The location of the sensors. The vision information of the participants’ motions is collected through cameras from three perspectives to assist in the classification. One sensor was placed in the center of the participant’s spine and another one on the right shoulder to collect data on the participant’s motions.

**Figure 2 bioengineering-10-00127-f002:**
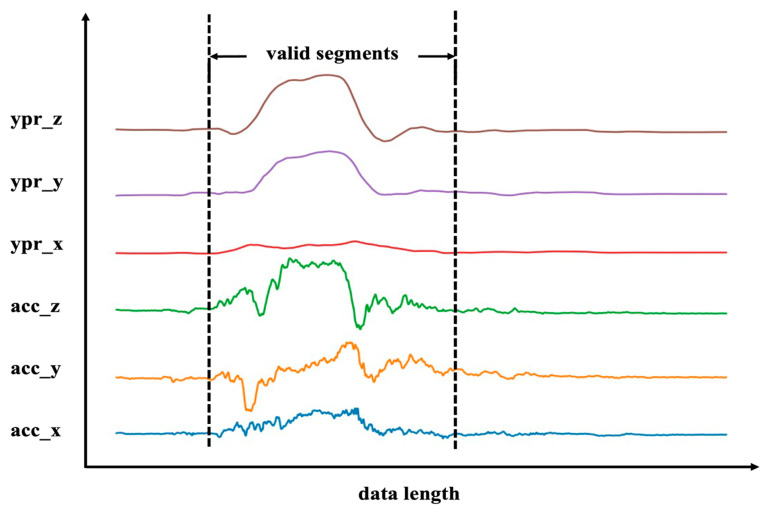
Take action 6 as an example to display the data of each channel of the back sensor. The lines from bottom to top represent the accelerometer *x*-axis(acc_x), *y*-axis(acc_y), *z*-axis(acc_z), gyroscope *x*-axis(ypr_x), *y*-axis(ypr_y), and *z*-axis(ypr_z). Valid segments of motions are shown within dashed lines.

**Figure 3 bioengineering-10-00127-f003:**
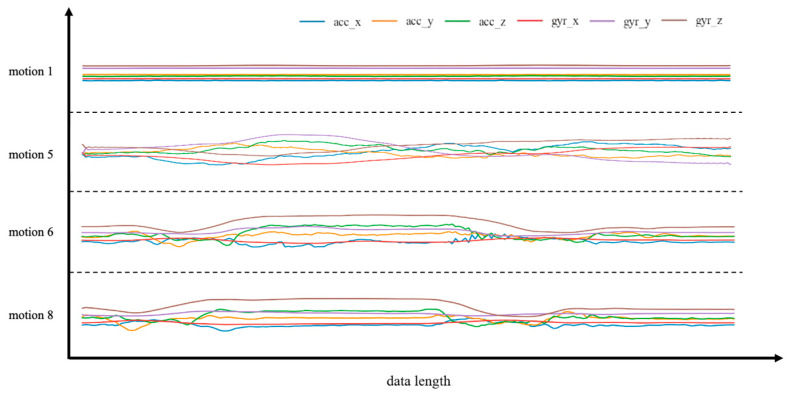
The randomly-selected four different actions of one of the participants and the data of the accelerometer and gyroscope data of the back sensor. The selected actions are as follows: motion 1 (sitting still), motion 5 (turning around and looking around), motion 6 (raising hand while standing up) and motion 8 (standing up and sitting down). We uniformly downsampled the data length to 200 for display clarity. Through motion sensors, we can continuously collect data about different motions, and each motion has a unique motion pattern. The relative intensity of each action is reflected in the ordinate after normalization.

**Figure 4 bioengineering-10-00127-f004:**
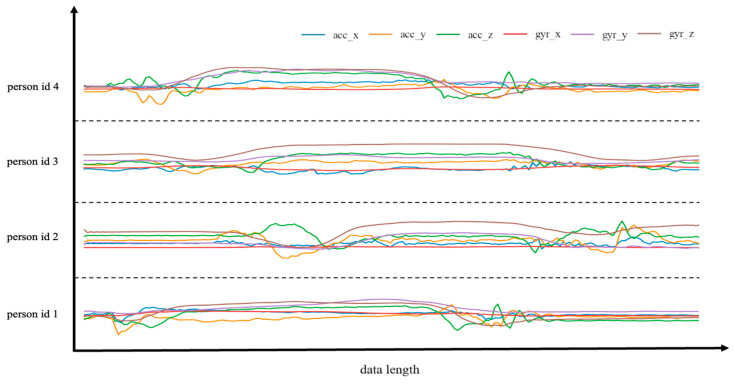
In the same action mode, the data of four volunteers (id1, id2, id3, id4) are randomly selected for display. We uniformly downsampled the data length to 200 for display clarity. It was challenging to classify classroom behavior since each participant carried out the same action in different ways and had unique sensor data patterns.

**Figure 5 bioengineering-10-00127-f005:**
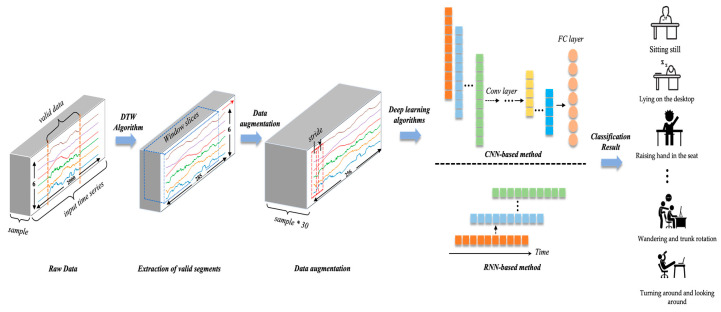
The framework of the whole process of the algorithm. The algorithm takes raw data as the input and outputs the most likely behavior from the 14 common classroom behaviors.

**Figure 6 bioengineering-10-00127-f006:**
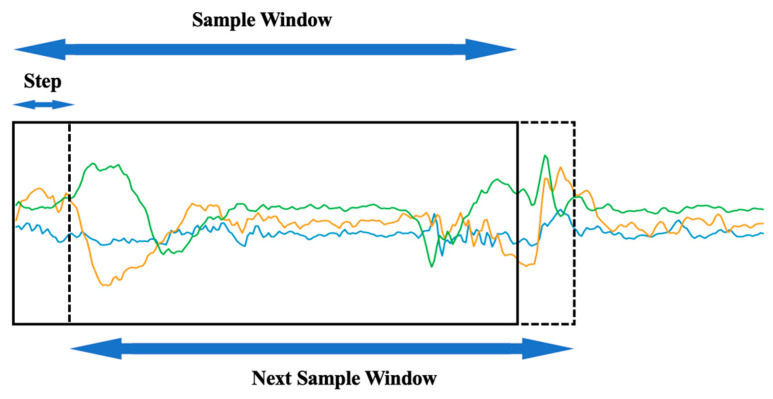
The detailed data augmentation procedure. The green/orange/blue lines represent the sensor data for a motion. The detailed data augmentation procedure. The green/orange/blue lines represent the sensor data for a motion. The sample window size is 256, and the stride size of the window is 1. We received 30 identical labeled data for each 285-length motion data after data augmentation.

**Figure 7 bioengineering-10-00127-f007:**
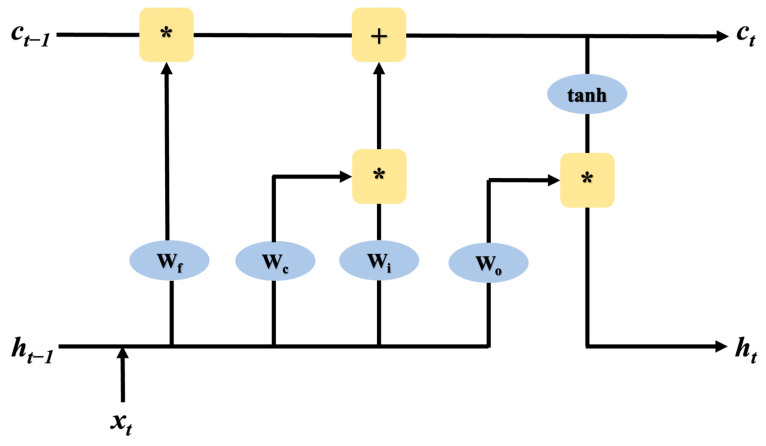
LSTM structure, W_f_ is the forgetting gate, W_i_ is the input gate, W_o_ is the output gate, x_t_ is the input data, h_t−1_ is the neural node of the hidden state, and W_f_ is used to calculate the features in c_t−1_ to obtain c_t_.

**Figure 8 bioengineering-10-00127-f008:**
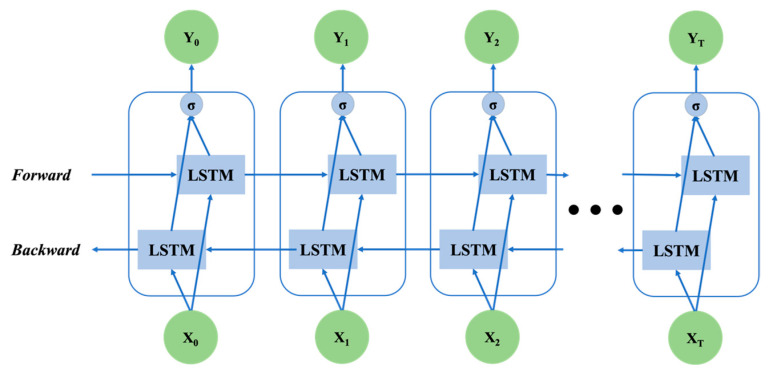
Bi-LSTM structure, which combines forward LSTM and backward LSTM.

**Figure 9 bioengineering-10-00127-f009:**
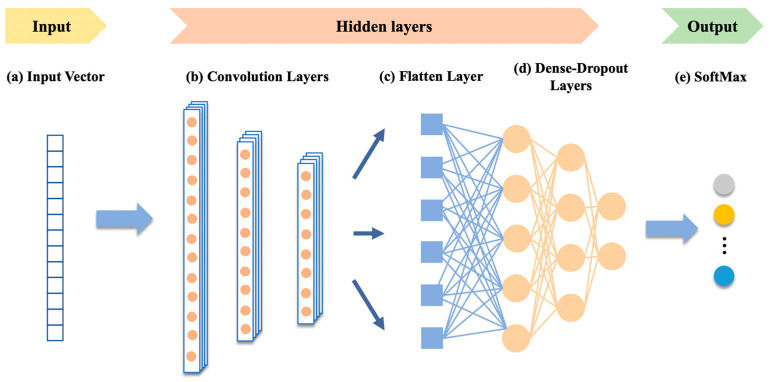
1DCNN structure. The structure of 1DCNN mainly includes input, hidden layer, and output, so as to achieve the purpose of feature extraction.

**Table 1 bioengineering-10-00127-t001:** Motion mode design. To simulate classroom behaviors for the participants, we selected 14 typical classroom behaviors. The table lists each motion’s name as well as the order in which it took place.

Serial No.	Motion Mode
1	Sitting still
2	Lying on the desktop
3	Writing notes
4	Raising a hand in the seat
5	Turning around and looking around
6	Raising a hand while standing up
7	Rocking on the seat
8	Standing up and sitting down
9	Wandering and trunk rotation
10	Playing hands
11	Turning pen in hand
12	Knocking on the desktop
13	Leaning the body and chatting
14	Shaking legs

**Table 2 bioengineering-10-00127-t002:** Jaccard index for 13 motions. All the extracted valid segments’ indices except lying on the desktop and writing notes are more than 88% similar to the benchmark.

Motion Mode	Jaccard Index
Raising a hand in the seat	0.97
Turning around and looking around	0.96
Raising hand while standing up	0.96
Rocking on the seat	0.97
Stand up and sit down	0.97
Wandering and trunk rotation	0.98
Playing hands	0.87
Turning pen in hand	0.88
Knocking on the desktop	0.95
Leaning the body and chat	0.96
Shaking legs	0.94
Lying on the desktop	0.45
Writing notes	0.50

**Table 3 bioengineering-10-00127-t003:** Main Result of four networks for the back sensor and shoulder sensor separately. Furthermore, acc represents accelerometer data, ypr represents gyroscope data, and acc + ypr represents the combination of accelerometer and gyroscope data.

	Back			Shoulder		
Accuracy (%)	acc	ypr	acc + ypr	acc	ypr	acc + ypr
DNN	81.8	91.2	93.3	89.5	86.5	91.7
LSTM	66.5	84	96.4	81.3	81.6	89.2
BiLSTM	96	98	99.8	96.4	95.9	97.2
1DCNN	99.8	99.9	100	99.6	98.3	98.8

**Table 4 bioengineering-10-00127-t004:** Test result of the effectiveness of VB-DTW valid segment extraction.

	With VB-DTW Valid Segment Extraction		Without VB-DTW Valid Segment Extraction		Improvement by VB-DTW	
Accuracy (%)	Back	Shoulder	Back	Shoulder	Back	Shoulder
DNN	93.3	91.7	89.6	82.5	3.7↑	9.2↑
LSTM	96.4	89.2	92.1	85.2	4.3↑	4.0↑
BiLSTM	99.8	97.2	93.8	90.2	5.0↑	7.0↑
1DCNN	100	98.8	98.5	95.9	1.5↑	2.9↑

**Table 5 bioengineering-10-00127-t005:** Test result of the effectiveness of data augmentation.

	With VB-DTW Augmentation		Without VB-DTW Augmentation		Improvement by VB-DTW	
Accuracy (%)	Back	Shoulder	Back	Shoulder	Back	Shoulder
DNN	93.3	91.7	41.2	39.0	52.1↑	52.7↑
LSTM	96.4	89.2	49.5	51.1	46.9↑	38.1↑
BiLSTM	99.8	97.2	52.8	51.1	47.0↑	46.1↑
1DCNN	100	98.8	53.8	52.1	46.2↑	46.7↑

## Data Availability

The datasets used and/or analyzed during the current study are available from the corresponding author upon reasonable request.

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
