# Peer review of "Automated Student Classroom Behaviors’ Perception and Identification Using Motion Sensors"

_bioengineering, 2023, doi:10.3390/bioengineering10020127_

Round 1
Reviewer 1 Report
The paper proposed a systematic design and development of an intelligent system for automated detection and identification of classroom students' behaviours from motion sensors. The proposed deep learning based model and algorithm were well designed and the experimental validations and ablation studies on their self-constructed dataset showed very impressive results. The introduction and research background were presented with comprehensive literature review and the method was logically elucidated. Overall the paper was presented in a well-structured and easy-to-follow manner. Please find some comments as below to address to further improve the quality and clarity of the paper for potential acceptance.
The title could be revised as "Automated Perception and Identification of Student Classroom Behaviors Using Motion Sensors".
Experimental comparisons with the SOTA methods/models would be helpful to further validate and justify the superior performance of the proposed method.
More detailed and insightful discussions would be helpful to improve the clarity and support better understanding on the underlying mechanism contributing to the impressive results. Further, it would be nice to see the discussion on any limitations of the proposed method.
Lastly, when overall presentation was quite good, careful proofreading would be necessary for a quality presentation.
Author Response
Comment 1: The paper proposed a systematic design and development of an intelligent system for automated detection and identification of classroom students' behaviors from motion sensors. The proposed deep learning-based model and algorithm were well designed and the experimental validations and ablation studies on their self-constructed dataset showed very impressive results. The introduction and research background were presented with comprehensive literature review and the method was logically elucidated. Overall, the paper was presented in a well-structured and easy-to-follow manner. Please find some comments as below to address to further improve the quality and clarity of the paper for potential acceptance.
Response: Thank you for the comment.
Comment 2: The title could be revised as "Automated Perception and Identification of Student Classroom Behaviors Using Motion Sensors".
Response: Thank you for the valuable suggestion. The title of the paper has been changed to "Automated Perception and Identification of Student Classroom Behaviors Using Motion Sensors".
Comment 3: Experimental comparisons with the SOTA methods/models would be helpful to validate further and justify the superior performance of the proposed method.
Response: Thanks for the comment. According to our knowledge, we are the first team to recognize students' behavior in the classroom via motion sensors. There are no existing methods and dataset specially for classroom behavior recognition. Therefore, we proposed a three-stage algorithm, including the extraction of valid segments by Dynamic Time Warping algorithm, data augmentation, and a deep learning-based classification model. In the comparison experiment of deep learning-based classification model, we compared four SOTA models (DNN, LSTM, BiLSTM, and 1DCNN) in AI field and chosen 1DCNN, the one with best performance. As a whole, the proposed method gets up to 100% accuracy on a self-built dataset.
Comment 4: More detailed and insightful discussions would be helpful to improve the clarity and support better understanding on the underlying mechanism contributing to the impressive results. Further, it would be nice to see the discussion on any limitations of the proposed method.
Response: Thank you for the valuable suggestion. A new section “4. Discussion” is added, which include ablation study and limitations of the proposed method. In the ablation study, the contributions of the components in the proposed method are investigated and discussed.
Comment 5: Lastly, when overall presentation was quite good, careful proofreading would be necessary for a quality presentation.
Response: Thank you for the valuable suggestion. The manuscript is revised using the “Track Changes” function. The revised manuscript has been edited by a proof-reader and the revised parts are highlighted.

Reviewer 2 Report
This study examined a motion sensor-based intelligent system's effectiveness in realising the perception and identification of classroom behaviour. The manuscript was well-written; however, some important information was lacking, which required some work before publication.
Abstract: Please explain the Voting-Based Dynamic Time Warping algorithm (VB-DTW). How does this function? Then, how is the sample collected? Is it representative of the population sample?
Introduction: The introduction needs more information on the importance of building intelligent digital education platforms. What is the significance of this study?
Methods:
Unfortunately, the sample size needs to be larger to achieve the power of the study. Please include sample size calculation. The age of the participants also is large. Please give justification.
Discussion – suggest separating results and discussion.
Please include the limitation of this study. All the best.
Author Response
Comment 1: This study examined a motion sensor-based intelligent system's effectiveness in realizing the perception and identification of classroom behavior. The manuscript was well-written; however, some important information was lacking, which required some work before publication.
Response: Thank you for the comment. The manuscript is revised using the “Track Changes” function.
Comment 2: Abstract: Please explain the Voting-Based Dynamic Time Warping algorithm (VB-DTW). How does this function? Then, how is the sample collected? Is it representative of the population sample?
Response: Thank you for the valuable suggestion. Explanation to the proposed VB-DTW algorithm has been added in the abstract. “For the acquired sensor signal, we proposed a Voting-Based Dynamic Time Warping algorithm (VB-DTW) in which a voting mechanism is used to compare the similarities between adjacent clips and extract valid action segments.”. The sample collection information has been added in the abstract. “To verify the feasibility of the proposed method, a self-constructed dataset (SCB-13) was collected. Thirteen participants were invited to perform 14 common class behaviors, wearing motion sensors whose data were recorded by a program.”
Comment 3: Introduction: The introduction needs more information on the importance of building intelligent digital education platforms. What is the significance of this study?
Response: Thanks for the comment. More information about the importance of the intelligent digital education platforms and how our work influences its construction has been added to the introduction.
Comment 4: Unfortunately, the sample size needs to be larger to achieve the power of the study. Please include sample size calculation. The age of the participants also is large. Please give justification.
Response: Thank you for the valuable suggestion. We recruited 13 participants for data collection. Each participant was required to perform 14 common class behaviors. Each behavior was repeated for 5 times. Therefore, we obtained 910 (13x14x5) samples. The result of deep network classification is not satisfactory when the 910 samples are utilized in training (please refer to the second part of the ablation study for details). Therefore, we proposed to use data augmentation to boost the samples. Data augmentation is a widely used strategy in machine learning that increases the training samples automatically, when data collection on a large population is difficult or infeasible. After data augmentation, the original data was increased by 20 times. That means 18,200 samples were generated, which is much larger than the minimum sample size for this research. The effectiveness of data augmentation is further verified in the ablation study.
This study aims at developing a system and its algorithms for classroom behavior recognition. The purpose of the data collection is to verify the effectiveness of the proposed algorithms. In the data collection, we requested the participants to imitate the classroom behavior. Therefore, there should be no significant differences in the actions between adults and children, under a controlled condition. The data collection and recognition on a larger group of school aged students will be conducted in the next stage of this study.
Comment 5: Discussion – suggest separating results and discussion. Please include the limitation of this study.
Response: Thank you for the valuable suggestion. The Discussion have been separated from the results, and the limitations of the paper are discussed in the second part of the Discussion.
